# Use of Autologous Platelet Rich Plasma (A-PRP) for Postpartum Perineal Repair Failure: A Case Report

**DOI:** 10.3390/jpm12111917

**Published:** 2022-11-17

**Authors:** Farida Akhoundova, Fanny Schumacher, Marie Léger, Sarah Berndt, Begoña Martinez de Tejada, Jasmine Abdulcadir

**Affiliations:** 1Department of Pediatrics, Gynecology and Obstetrics, Geneva University Hospitals, 1205 Geneva, Switzerland; 2Department of Plastic, Reconstructive and Aesthetic Surgery, Faculty of Medicine, Geneva University Hospitals, 1205 Geneva, Switzerland; 3Regen Lab SA, 1052 Le Mont-sur-Lausanne, Switzerland

**Keywords:** wound dehiscence, postpartum complication, perineal tear, episiotomy, autologous platelet-rich plasma (A-PRP), wound cicatrization

## Abstract

Perineal wound dehiscence is an uncommon but important postpartum complication. In many cases, it leads to extreme pain and urinary and defecation problems. For up to several weeks, it can interfere with the mother’s daily activity, affecting psychosexual health and body image. The best way to manage perineal wound breakdown (resuturing vs. spontaneous closure) after childbirth remains controversial. A-PRP is the autologous human plasma containing an increased platelet concentration, rich in growth factors, and mediators with hemostatic, anti-inflammatory, and antimicrobial properties. It accelerates the natural healing process. Even though A-PRP is widely used in orthopedics and dermatology, its use in gynecological injuries is limited. We describe here a case of a woman with postpartum perineal dehiscence treated with A-PRP with positive outcomes.

## 1. Background

Autologous platelet-rich plasma (A-PRP) is an innovative and promising treatment for a large spectrum of soft tissue injuries. A-PRP is widely used in orthopedics for musculoskeletal disease and dermatology for the treatment of ulcers, scars, and alopecia [1].

There is limited experience with the use of A-PRP to treat gynecological injuries, and no studies reporting the administration of A-PRP to improve the treatment of obstetric perineal repair failure have been published [2,3].

We report a case of a young woman who underwent A-PRP injections as part of her episiotomy dehiscence treatment.

## 2. Case Presentation

A 28-year-old woman with no relevant medical history had a vacuum-assisted delivery with a right mediolateral episiotomy and a second-degree tear. Her delivery was complicated by the severe postpartum hemorrhage of 1 L due to extensive vaginal varicosities and mild uterine atony, successfully managed with 25 UI of oxytocin and 1 g of intravenous (IV) tranexamic acid.

The perineal laceration was repaired using a running unlocked suture with a standard-absorbable 2/0 polyglactin (for the mucosal and muscle layers) ending with intradermal skin closure. The suture was performed by a senior gynecologist. On day 4 post-partum, our patient was discharged after showing adequate perineal healing and after receiving a 1 g iron infusion because of moderate anemia.

On day 7 post-partum, the woman came to our emergency unit complaining of fever, shivering, increased yellow vaginal discharge, and worsened perineal pain that started 48 h before. She was unable to sit or walk. Her heart rate was 120 bpm and her temperature was 38.2 °C. An examination showed a non-infected 2 cm deep and 3 cm long dehiscence, including the muscle layer, without fistulization towards the rectum (Figure 1A); uterine pain and abnormal vaginal discharge were suspicious for postpartum endometritis. A blood test revealed leukocytosis (14 × 10^3^ cells/L) with a C-reactive protein level of 40 mg/L, a hemoglobin level of 8.8 g/dL, and a platelet count of 381 × 10^3^/µL. The patient was re-admitted for local conservative management (irrigation) of the perineal dehiscence; amoxicillin/clavulanic acid at 2 g/6 h IV was provided for the endometritis, and acetaminophen, ibuprofen, and oral morphine were used as analgesics.

On day 11, the patient was still in pain and a discussion about the benefits and risks of re-suturing was conducted. The patient preferred a conservative approach (wound irrigation, sterile medical honey application) with the administration of A-PRP.

One single treatment of 10 mL of autologous fibrin glue (PRP-activated with autologous thrombin serum (ATS); RegenKit-Surgery of Regen Lab^®^ SA, Le Mont-sur-Lausanne, Switzerland) was administered in the perineal wound (edges and bed) as an office procedure without anesthesia (Figure 1B). Figure 2 shows a detailed description of the A-PRP fibrin glue preparation process.

She was discharged 2 days later with 10 more days of oral amoxicillin/clavulanic acid at 1 g/8 h for endometritis, which had an uneventful evolution.

Perineal care with sitz baths, daily wound application of sterile honey, analgesia with nonsteroidal anti-inflammatory drugs and acetaminophen, and osmotic laxatives was prescribed.

A weekly outpatient follow-up visit with a vulva specialist was organized.

## 3. Discussion

Our case illustrates a non-standard therapeutic approach of A-PRP administration in a woman with vaginal wound dehiscence after an assisted delivery with successful results. Complete healing of the perineal wound took three weeks with a gradual reduction in the pain, which was solved by day 30 (Figure 1C,D). This outcome allowed her the possibility of continuing to breastfeed as well as being able to start pelvic floor physiotherapy two months after childbirth. Perineal wound dehiscence is an uncommon but important postpartum complication. It usually occurs during the first 7–14 days after childbirth with a wide variation in prevalence (from 0.59% to 13.5% [4]). Its major risk factors are operative delivery and episiotomy, as in our patient; severe obesity (body mass index >35 kg/m^2^); infection; and third- or fourth-degree laceration [5,6,7].

In many cases, perineal wound failure is associated with extreme pain and urinary and defecation problems for up to several weeks, thus interfering with the mother’s daily activity. Moreover, concerns about poor healing and altered body image can lead to psychosexual morbidity, which could last up to 9 months [4,6,7,8].

The best way to manage perineal wound breakdown (re-suturing vs. spontaneous closure) after childbirth remains unclear due to a lack of conclusive evidence regarding the outcomes of both management options, such as perineal pain, time to heal, or dyspareunia.

In clinical experience as well as the literature [8], most women with first- or second-degree perineal injuries do well with expectant management. Nevertheless, healing can be lengthy and difficult to bear due to persistent perineal discomfort or pain, concerns about final functionality and aesthetic results, and the long follow-up. These factors may cause an overwhelming desire for a quick treatment option, resulting in some women and health professionals choosing surgical management [4].

A recent retrospective study [9] supported early re-suturing (i.e., within 14 days after vaginal delivery [7]) of perineal wound dehiscence due to a faster healing time, reduced follow-up requirements, and few major complications.

Two other retrospective studies and a few RCTs showed that healing time was around 2–3 weeks for resutured episiotomies [10,11] compared to those that were not resutured, which experienced longer healing times (>4 weeks) [12].

However, resuturing is not without risks. The stress of surgical debridement and the possibility of an early close attempt failure may not be favored by all women. Readmission or a prolonged hospital stay may disrupt breastfeeding and maternal–infant bonding.

Both conservative management and surgical repair can result in poor healing and eventual further corrective surgery, perineal refashioning, or the excision of excessive scar or granulation tissue formation [4].

After more than two decades of challenging the management of perineal wound dehiscence [13], the need for a therapeutic alternative that can shorten the healing of this complication remains needed.

A-PRP is a preparation of autologous human plasma containing an increased platelet concentration, and it is rich in growth factors and mediators with hemostatic and healing properties. Platelet alpha-granules contain more than 300 bioactive substances [14]. These substances play a fundamental role in hemostasis and/or tissue healing through the stimulation of cellular chemotaxis, proliferation, and differentiation; the removal of tissue debris; angiogenesis; and the deposition of the extracellular matrix [15]. Among them are notable growth factors involved in the healing process, such as transforming growth factor beta (TGF-β), vascular endothelial growth factor (VEGF), platelet-derived growth factor (PDGF), epithelial growth factor (EGF), basic fibroblast growth factor (bFGF), keratinocyte growth factor (KGF), connective tissue growth factor (CTGF), and insulin-like growth factors (IGF). Some of them, such as PDGF, have a paracrine effect on different cell types, such as mesenchymal stem cell recruitment [16]. Platelets also release histamine and serotonin, which increases local capillary permeability to improve access for additional inflammatory cells to start the reparative process [17].

This slightly supraphysiologic amount of growth factors released at the injury site accelerates the natural healing process [2]. Its anti-inflammatory and antimicrobial properties contribute to pain relief and the prevention of infection [18,19]. The topical application of A-PRP has been tested in complex cases of chronic immunosuppression, infected wounds, and after ineffective vacuum-assisted closure therapy with positive outcomes [20]. Furthermore, A-PRP has improved re-epithelization in acute wounds, such as acute ulcers, wound dehiscence following surgery, and burns. It may achieve up to 80–100% of epithelization between 4 and 20 days [21]. The treatment procedure entails non-invasive administration using minimally manipulated autologous platelet concentrate, thereby decreasing the chances of infection and cross-contamination while processing and administering the cellular therapy [22].

For our patient, the A-PRP solution was prepared directly in the examination room of the hospital in less than 15 min.

Our case shows that A-PRP may be an option for expediting and improving the conservative management of the dehisced wound after vaginal delivery. Administration in an outpatient setting, not needing anesthesia, and there being no major adverse effects [19] make this therapy a promising option.

However, some contraindications do exist. Absolute contraindications are patients with hereditary or acquired hematologic/coagulation disorders or patients suffering from severe metabolic or systemic disorders. Some relative contraindications are malignancy; auto-immune diseases (Hashimoto, rheumatoid arthritis, lupus, etc.); patients who have taken, within 3 days, aspirin or other medications that alter platelet function; recent fever, illness, or anemia (HGB < 10 g/dL).

A recent publication has reviewed the various research works on the use of platelet-rich plasma in gynecology [23]. Various scientific observations were obtained, which have the potential to bring about a great revolution in obstetrics and gynecology. Starting from very minor conditions to the most chronic forms of certain disorders, platelet-rich plasma is undeniably a promising candidate. Indeed, promising clinical data are arising for thin endometrium, recurrent genital fistulas, ovarian abnormalities or poor ovarian reserve, Asherman’s syndrome, urinary stress incontinence, etc. [23].

Further research is needed to provide robust and long-term evidence of the efficacy of A-PRP in perineal dehisced obstetric wounds.

## Figures and Tables

**Figure 1 jpm-12-01917-f001:**
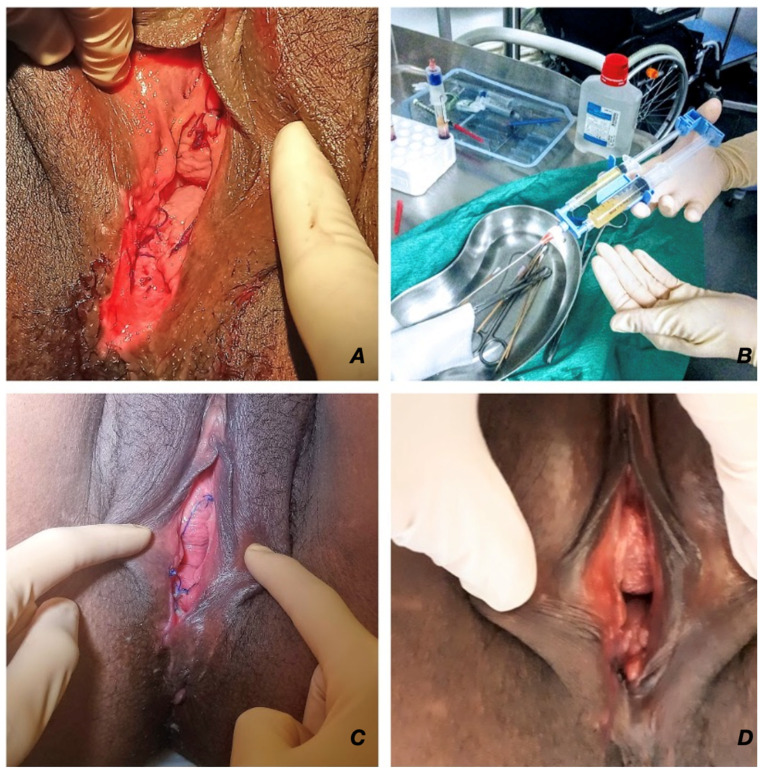
(**A**) Perineal status on the day of admission: a non-infected breakdown of a 2 cm deep and 3 cm long dehiscence, including the muscle layer. (**B**) Autologous fibrin glue preparation obtained from RegenKit-Surgery (Regen Lab^®^ SA, Switzerland). (**C**) Day 7 following the A-PRP administration. (**D**) Day 22 following the A-PRP administration.

**Figure 2 jpm-12-01917-f002:**
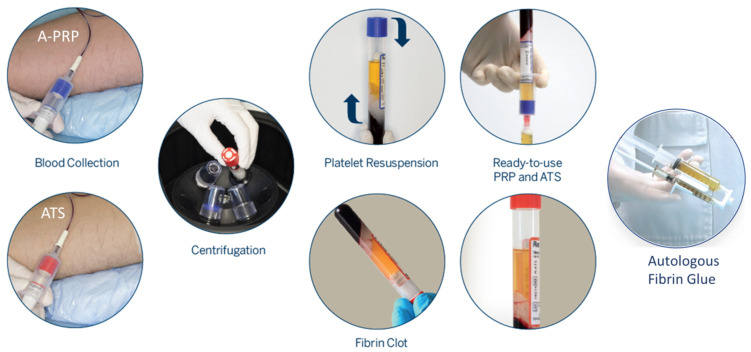
Preparation of autologous fibrin glue using Regenkit-Surgery, RegenLab Switzerland. The kit allows the close circuit preparation of A-PRP and ATS using a separating gel technology. The kit contains two tubes with sodium citrate anticoagulant (blue cap) for PRP preparations and one tube without anticoagulant (red cap) for autologous thrombin preparation (ATS). The venous puncture is performed, and the three tubes are filled with whole blood. The vacuum within the tubes enables automatic collection of the necessary volume of blood (about 10 mL). Tubes are centrifuged to allow blood component separation. After centrifugation, the blood is fractionated; the red and white blood cells are trapped under the separating gel, and platelets settle on the surface of the gel. The tube is gently rocked to resuspend the platelets and about 5 mL of PRP will be obtained for each tube.During centrifugation of RegenATS tube, a clot of fibrin settles on the separating gel. The liquid part that is extracted from the clot constitutes the activated thrombin serum which is used to trigger the coagulation of anticoagulated PRP. PRP from the two tubes is collected in a 10 mL syringe. Two mL of autologous thrombin serum are collected in a 3 mL syringe in which 1 mL calcium gluconate (10% injectable) is added. Then the 2 syringes are mounted on the RegenSpray applicator and the preparation is ready for use on the patient. Using the applicator allows to combine the PRP and its activators at the time of application.

## Data Availability

Not applicable.

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
