# Peer review of "Use of Autologous Platelet Rich Plasma (A-PRP) for Postpartum Perineal Repair Failure: A Case Report"

_jpm, 2022, doi:10.3390/jpm12111917_

Round 1

Reviewer 1 Report

Although the preparation is not a novelty, as the authors also say, the use of A-PRP in the treatment of a dehiscent postpartum perineal wound represents a new direction of successful use. From this point of view, the case study deserves to be published

Author Response

Comments and Suggestions for Authors

Although the preparation is not a novelty, as the authors also say, the use of A-PRP in the treatment of a dehiscent postpartum perineal wound represents a new direction of successful use. From this point of view, the case study deserves to be published

We thank the reviewer for this kind comment.

Reviewer 2 Report

The authors presented a case where A-PRP injections were used as part of perineal dehiscence treatment.

This is a very interesting case for several reasons: firstly, peripartal perineal ruptures of various degrees sometimes, and not so rarely, require an experienced and senior obstetrician for complete reconstruction and they could cause significant blood loss. Furthermore, even after reconstruction, poor wound healing and injuries themselves could cause persistent perineal pain, and dyspareunia inevitably leading to a major psychological burden. On the other hand, the usage of PRP in gynecology is so far described, according to the literature, mostly in the treatment of poor ovarian reserve and implantation failure. Therefore, combined with the significance of perineal tears and dehiscences, I find the described approach of treatment with A-PRP interesting and significant. 

The literature is up-to-date, the images are of high quality, and the English language needs a minor spell check.

However, I would encourage authors to include a bit more detailed description of the A-PRP preparation process. I understand that there is a lack of standardization in preparation and that the preparation depends on the manufacturer’s instruction, but for the sake of future studies, a brief step-by-step preparation process would improve the scientific soundness of the paper. Furthermore, the Discussion section would benefit from the inclusion of a paragraph about the current knowledge of A-PRP in gynecological practice.

Here are some specific comments:

1. Case presentation, line 42: „On day 7“: Should be changed to „…day 7 post-partum“

2. Case presentation, lines 46, 57, and 65: You mention Figure 1 (A, B, C-D) but the uploaded figure is named „Figure 2“. Is there another Figure that you forgot to upload or is this a miss-spell? You should check this out.

3. Case presentation, line 64: „Complete healing of the perineal wound took 3 weeks“: You should use this information and further discuss it in the Discussion section: time to wound healing: suturing (or, as you mentioned, early suturing) vs. A-PRP. I understand that there is limited data to be compared, but for usage in future systematic reviews or meta-analyses, you should consider including this information. 

4. Discussion, paragraph 9: I encourage you to provide a bit more-detailed characteristics of platelet growth factors. 

5. Discussion: You should include contraindications of A-PRP usage. 

Author Response

The authors presented a case where A-PRP injections were used as part of perineal dehiscence treatment.

This is a very interesting case for several reasons: firstly, peripartal perineal ruptures of various degrees sometimes, and not so rarely, require an experienced and senior obstetrician for complete reconstruction and they could cause significant blood loss. Furthermore, even after reconstruction, poor wound healing and injuries themselves could cause persistent perineal pain, and dyspareunia inevitably leading to a major psychological burden. On the other hand, the usage of PRP in gynecology is so far described, according to the literature, mostly in the treatment of poor ovarian reserve and implantation failure. Therefore, combined with the significance of perineal tears and dehiscences, I find the described approach of treatment with A-PRP interesting and significant. 

The literature is up-to-date, the images are of high quality, and the English language needs a minor spell check.

We thank the reviewer for this kind comment.

However, I would encourage authors to include a bit more detailed description of the A-PRP preparation process. I understand that there is a lack of standardization in preparation and that the preparation depends on the manufacturer’s instruction, but for the sake of future studies, a brief step-by-step preparation process would improve the scientific soundness of the paper.

For this comment, we propose to add the Figure 1 that describes the flowchart of the PRP preparation.

Figure 1. Preparation of autologous fibrin glue using Regenkit-Surgery, RegenLab Switzerland.

The kit allows the close circuit preparation of A-PRP and ATS using a separating gel technology. The kit contains two tubes with sodium citrate anticoagulant (blue cap) for PRP preparations and one tube without anticoagulant (red cap) for autologous thrombin preparation (ATS). The venous puncture is performed, and the three tubes are filled with whole blood. The vacuum within the tubes enables automatic collection of the necessary volume of blood (about 10 ml). Tubes are centrifuged to allow blood component separation.  After centrifugation, the blood is fractionated; the red and white blood cells are trapped under the separating gel, and platelets settle on the surface of the gel.  The tube is gently rocked to resuspend the platelets and about 5 ml of PRP will be obtained for each tube. During centrifugation of RegenATS tube, a clot of fibrin settles on the separating gel. The liquid part that is extracted from the clot constitutes the activated thrombin serum which is used to trigger the coagulation of anticoagulated PRP.PRP from the two tubes is collected in a 10 ml syringe. Two ml of autologous thrombin serum is collected in a 3 ml syringe in which 1 ml calcium gluconate (10% injectable) is added. Then the 2 syringes are mounted on the RegenSpray applicator and the preparation is ready for use on the patient. Using the applicator allow to combine the PRP and its activators at the time of application.

.

Furthermore, the Discussion section would benefit from the inclusion of a paragraph about the current knowledge of A-PRP in gynecological practice.

This is an important point, thank you, we have now added this paragraph:

A recent publication has carefully reviewed the various research works on the use of platelet-rich plasma in gynecology, published on internationally recognized scientific platforms. Various scientific observations were obtained, which have the potential to bring about a great revolution in obstetrics and gynecology. Starting from very minor conditions to the the most chronic forms of disorders, platelet-rich plasma is undeniably a promising candidate. Indeed, promising clinical data are arising for thin endometrium, recurrent genital fistulas, ovarian abnormalities or poor ovarian reserve, Asherman’s syndrome, urinary stress incontinence, etc., [Varghese et al. 2022].

Here are some specific comments:

  1. Case presentation, line 42: „On day 7“: Should be changed to „…day 7 post-partum“

This is now corrected in the new version.

  1. Case presentation, lines 46, 57, and 65: You mention Figure 1 (A, B, C-D) but the uploaded figure is named „Figure 2“. Is there another Figure that you forgot to upload or is this a miss-spell? You should check this out.

In the submitted version, it was the only Figure, but as we know propose to add the PRP preparation process (Figure 1), the clinical pictures will be now Figure 2.

  1. Case presentation, line 64: „Complete healing of the perineal wound took 3 weeks“: You should use this information and further discuss it in the Discussion section: time to wound healing: suturing (or, as you mentioned, early suturing) vs. A-PRP. I understand that there is limited data to be compared, but for usage in future systematic reviews or meta-analyses, you should consider including this information. 

We now moved line 64 to “Discussion part” and added information about healing outcomes as recommended:

“Our case illustrates a non-standard therapeutic approach using A-PRP administration in a woman with a vaginal wound dehiscence after an assisted delivery with successful results. Complete healing of the perineal wound took three weeks with a gradual reduction of the pain, solved at day 30 (Figure 2C-D). That let her the possibility to continue breastfeeding and be able to start pelvic floor physiotherapy two months after childbirth.

…..“Two other retrospective studies and little RCT showed that healing time was around 2–3 weeks in resutured episiotomies [Ramin et al. 1992, Uygur et al. 2004] compared to these that were not resutured, which experienced longer healing times (>4 weeks) [Christensen et al 1994].”

  1. Discussion, paragraph 9: I encourage you to provide a bit more-detailed characteristics of platelet growth factors. 

We have now added this paragraph:

Platelet alpha-granules contain more than 300 bioactive substances [Golebiwska et al. 2015]. These substances play a fundamental role in hemostasis and/or tissue healing by stimulating cellular chemotaxis, proliferation and differentiation, removal of tissue debris, angiogenesis,

and the deposition of extracellular matrix [Mehta et al. 2008]. Among them are notable growth factors involved in the healing process such as: transforming growth factor beta (TGF-β), vascular endothelial growth factor (VEGF), platelet-derived growth factor (PDGF),

epithelial growth factor (EGF), basic fibroblast growth factor (bFGF) keratinocyte growth factor (KGF), connective tissue growth factor (CTGF) and insulin-like growth factors (IGF).

Some of them, like PDGF, have a paracrine effect on different cell types like mesenchymal stem cells recruitment [Wu et al. 2016]. Platelets also release histamine and serotonin, which increases local capillary permeability to improve access for additional inflammatory cells to start the reparative process [Degen, 2017].

  1. Discussion: You should include contraindications of A-PRP usage. 

We have now added this paragraph:

However, some contraindications do exist. Absolute contraindications are: patients with hereditary or acquired hematologic/coagulation disorders or patients suffering from severe metabolic or systemic disorders.

Some relative contraindications are malignancy, auto-immune diseases (Hashimoto, rheumatoid arthritis, lupus, etc.), patients who have taken, within 3 days, aspirin or other medications that alter platelet function, recent fever or illness or anemia (HGB <10g/dl).
